# Smart Cities Mission in the Face of COVID: Scope and Scale of ‘Smart’ COVID Responses in India

**DOI:** 10.3390/ijerph20227036

**Published:** 2023-11-08

**Authors:** Tooran Alizadeh, Lizwin Kurian, Chakshu Bansal, Deepti Prasad

**Affiliations:** School of Architecture Design and Planning, The University of Sydney, Sydney 2007, Australiachakshu@sydney.edu.au (C.B.);

**Keywords:** COVID-19, smart technologies, subaltern, CovTech, smart cities, India

## Abstract

COVID has expedited and expanded the already precarious smart city development in India with the multitude of ‘smart’ COVID responses—broadly known as CovTech—introduced since the early days of the global pandemic. This paper offers an analysis of the scope and scale of smart responses to COVID in the first 20 cities prioritized for smart city implementation—as part of the Smart Cities Mission in India. The analysis unravels the diversity within the smart COVID response, as 125 COVID applications, 14 COVID-War-Rooms, and numerous examples of smart public place initiatives are discussed. The findings include a typology of COVID applications and shed light on the operations of COVID-War-Rooms throughout the nation. The learnings point toward a mostly top-down approach to smart COVID response. Yet, early evidence indicates the existence of an alternative subaltern smart COVID response to provide bottom-up support to the most vulnerable groups, filling the gaps in the top-down approach. More research is required to thoroughly understand the scope and scale of the subaltern smart response to COVID.

## 1. Introduction

In 2015, the Government of India created the Smart Cities Mission [1], which proposed the development of 100 smart cities to address the challenges of urbanization through smart solutions. This announcement was followed by categorizing the proposed 100 cities into five priority rounds for implementation. Smart city development, via the Smart Cities Mission in India, is broadly categorized as (1) Area-Based Development and (2) Pan City Development. Area-Based Development includes a range of retrofitting, redevelopment, and green-field projects such as major ICT clusters (to enable ‘smart’ employment), housing development (to support the ICT sector), place-making projects (e.g., urban refurbishment using ‘smart furniture’), and infrastructure projects (e.g., smart water and waste management facilities). Pan City Development covers city-wide initiatives that focus on technology-enabled urban management and mainly operates in the form of an Integrated-Command-and-Control-Centre.

During the global pandemic, India has recorded the world’s second highest number of COVID cases nationally (right after the United States). COVID has pushed Indian cities to reshape their smart city efforts and leverage smart solutions and facilities implemented as part of the Smart Cities Mission in the fight against the pandemic. For example, 47 cities across the country converted their Integrated-Command-and-Control-Centers to COVID-War-Rooms to lead the city-level emergency response. Proponents such as ‘India Smart Cities’ COVID Response’ [2] highlight that the tactical use of the Integrated-Command-and-Control-Centers coupled with data analytics enabled the administration to put in place a disaster response and recovery strategy. It is noted [3] that the central government’s smart cities program enabled a proactive approach through evidence-based urban governance, as the smart cities across the country took the pandemic as an opportunity to innovate, learn, collaborate, and find ways to respond to the crisis. Critics, on the other hand, have raised concerns, noting that CovTech—as public-sector-led technologies for the monitoring, management, and containment of the virus—violate data privacy protocols, are opaque [4], and can be used as surveillance tools [5] by the government with devastating economic and social impacts on marginalized groups [6]. 

In this paper, we argue that India’s ‘smart’ response to the global pandemic is multiscale and has different modes of existence, and further empirical studies are required to capture its complex socio-spatial implications. Section 2 starts with a brief literature review representing the CovTech challenges in India and beyond. Section 3 details the methodology describing the desktop analysis of the smart responses to COVID in the first 20 cities prioritized for smart city implementation as part of the Smart Cities Mission. The desktop analysis is complemented by learnings from our interviews with relevant NGOs across India. In Section 4, the findings include a typology of COVID applications and shed light on the mostly top-down approach to smart COVID response. Yet, early evidence indicates the existence of an alternative subaltern smart COVID response to provide bottom-up support to the most vulnerable groups, filling the gaps in the top-down approach. More research is required to thoroughly understand the scale and scope of the subaltern smart response to COVID. 

More broadly, a critical investigation of the smart responses to COVID sheds light on the opportunities and challenges embedded in smart urbanism in response to crises—whether economic, environmental, or otherwise.

## 2. Smart COVID Responses: A Brief Literature Review

A significant number of smart initiatives were adopted in response to COVID across the world to enable the tracing of patients and the spread of virus [7], homeschooling [8], working from home [9], and telemedicine/health [10], to name but a few. However, questions remain about their reach and impact towards the most vulnerable sections of society [5,11]. Moreover, research has not been able to fully capture the scope and scale of the smart responses to COVID. This is understandable, considering the magnitude of the global pandemic and the disruptions it caused in people’s livelihoods, including the researchers. Further, the existing research on the subject has been mostly written in a time when travel restrictions and COVID lockdowns were common, and the world had not embraced the new COVID normal. As a result, the research produced in 2020–2021 (and published all through 2022) has a higher-than-usual level of reliance on secondary sources and grey literature. The dominance of local news articles and unverified online sources reflects the COVID restrictions imposed on empirical work (e.g., site visits). With all these research gaps and limitations in mind, below we share what has so far been documented, with specific attention to CovTech in India—as the core of this paper.

A number of COVID applications—especially those introduced at the specific Indian city and state levels—have been reviewed in previous studies [4,6]. Examples include Quarantine Watch, COVA Punjab, Safe Kashi, and Saiyam; and the role these applications played in providing context-specific information in support of the local health systems has been acknowledged. Nevertheless, Goel et al. [12] point out the major issues faced in the absence of robust data exchange mechanisms, as COVID applications mostly worked in isolation (with limited or no data-sharing protocols) and did not allow for Centre-State-City coordination. 

In addition to the lack of coordination between the numerous COVID applications introduced by different levels of government in India, researchers—alongside civil society—have also raised concerns about breach of privacy and lack of transparency over the design and operation of such applications [13,14]. Privacy concerns are raised as the data collected via COVID applications could be used by the government to track the movement of people and their contacts beyond the public health mandate [15]. More specifically, the introduction of the selfie function in some of the quarantine apps is interpreted (and criticized) as an aspiration to extend the reach of state surveillance into intimate governance, with questions asked about who has access to such data apart from the health officials [4,16]. 

Beyond the privacy concerns, questions have been asked with reference to the pre-existing digital divide, which truly limits the reach of smart COVID responses. Such questioning is not limited to India; for example, Das and Zhang [17], focusing on Singapore, note that the pandemic has cruelly exposed the vulnerabilities of disadvantaged groups, especially in terms of effective engagement with ICTs, and minimized the chances of inclusion of the marginalized peripheral residents (such as informal dwellers and migrant communities) during the pandemic management in smart cities. In the context of India, the failure of COVID responses (smart or otherwise) to reach out to the most vulnerable sections of society was evident when horrifying images of immigrant laborers walking across the interstate highways hit the world media. As Raju et al. [18] point out, the failure to properly include immigrant laborers in the official COVID responses left them with no other option than to start walking. Immigrant laborers, mostly living in informal settlements, depended on daily labor and could not afford the luxury of working from home. Even though support services—e.g., the Prime Minister’s Citizen Assistance and Relief in Emergency Situations Fund—were provided, they were slow and, in many cases, excluded the most vulnerable people due to a lack of ration cards or proof of identity. 

## 3. Our Study

To understand the scope and scale of smart COVID responses in India, we focus on the first 20 cities prioritized for smart city implementation—as part of the Smart Cities Mission in India. This allows an investigation of smart COVID responses across the Indian cities positioned at the forefront of CovTech adoption due to their pre-COVID smart city investment. Figure 1 offers a spatial overview of the 20 smart cities investigated at the core of this paper.

We ran a desktop analysis to gather data on the diversity of smart COVID responses adopted in the 20 selected cities. The desktop investigation covers a variety of publicly available online sources, including (limited) academic research, official government websites and reports, media articles, social media posts, blogs, and app stores. As a result, we identified 125 COVID applications, 14 COVID-War-Rooms, and numerous examples of smart public place initiatives relevant to the 20 selected cities. More information and descriptions of the different categories of smart COVID responses are offered in the findings section of this paper. The learnings from the desktop analysis were complemented with preliminary interviews. Twenty NGO representatives from across the nation were interviewed to share their experience helping the most marginalized communities during COVID. The NGO selection focused on those vocal and active on the smart city front to capture civil society’s take on the smart COVID responses and understand the ways in which bottom-up initiatives were put in place to fill the gaps. Table 1 summarizes the proposed approach of this study.

Nevertheless, it is important to understand the challenges involved with this desktop analysis. Out of the 125 COVID applications identified, only 17 were previously noted in the academic literature. This means that—in line with the issues discussed in the brief literature review at the beginning of this paper—we had to rely heavily on grey literature and non-traditional data sources in our research. In response, we used triangulation of data and cross-examined multiple data sources to ensure the reliability and validity of the analysis. In a number of cases, the NGO representatives interviewed shared their local knowledge and networks on the ground to verify the discrepancies in the online data sources. Further, the desktop research, while shedding light on the scope and scale of smart COVID responses, is not able to fully determine the effectiveness of the variety of initiatives identified. Our complementary interviews with the relevant NGOs offer some answers to such questions. Nevertheless, more empirical research is needed to capture the complex socio-spatial implications [19] of such responses across different geographies and at different points in time throughout the pandemic.

## 4. Findings

### 4.1. Categories of CovTech

After the Indian government announced a complete lockdown on 24 March 2020, due to the outbreak of COVID-19, several smart initiatives were announced to keep the public informed and to monitor and contain the spread of the virus. In response to the global pandemic, different levels of government in India deployed smart technologies as a mode of ‘technological solutionism’ [5], broadly labelled as CovTech [20], supported by the infrastructures of apps, war rooms, and maps [6].

In this paper, we have classified the wide range of smart COVID responses in the 20 prioritized Indian smart cities under four categories: (1) COVID applications, (2) COVID-War-Rooms, (3) COVID smart public place initiatives, and (4) COVID smart subaltern responses. Below, we briefly define each CovTech category and share learnings from them.

#### 4.1.1. COVID Applications

COVID applications refer to a broad set of COVID-related digital platforms, including mobile-based apps, web-based apps, and helplines used to control and monitor the pandemic, and/or to provide medical and essential services to the people during the pandemic. Not all COVID applications were new; some were updated versions of pre-COVID applications reshaped in response to the crisis, such as the MyGov app. Below, we offer a typology of COVID applications to better understand their diversity.

We have identified 125 COVID applications operating within the 20 smart cities and categorized them based on their scale, technology dependency, and functionality. In terms of the scale, out of the 125 COVID applications, 63 are city-level, 48 are state-level, and 14 operate at the national level. From a technology dependency perspective, COVID applications are either web-based or mobile-based, and may be smartphone-dependent, Internet-dependent, or others (more information follows). In terms of functionality, the apps are categorized into 4 categories and 19 subcategories to provide a broad understanding of the scope and the diversity of services performed. 

Nevertheless, it is to be made clear that COVID applications are not necessarily dedicated to one function only and could be multifunctional. For example, the Government of India developed Aarogya Setu, the most popular COVID application in India, as a contact tracing app, which was then transformed into a national health app providing a one-stop solution. Aarogya Setu also provided information on COVID updates concerning each state [21], allowed to schedule and consult online doctor appointments, and also stored medical reports, including vaccination certificates [22]. Table 2 offers a summary of COVID applications by function and scale. 

The first broad functional category of COVID application, namely ‘General Information’, is the most popular category (*n* = 80), regardless of the scale of operation. ‘General Information’ is further divided into three functional subcategories, namely, ‘Information’, ‘Geospatial data’, and ‘Survey’. ‘Information’ apps provide general information related to COVID-19, such as the number of active and recovered cases. For example, Corona Safe Network in Kerala was a web-based app providing the daily status of COVID cases across the state and its district [23]. ‘Geospatial data’ apps provide spatial information (e.g., maps). Jabalpur Smart Surveillance COVID platform (JSS- COVID), for example, had an integrated GIS map spatially showing the location and containment zones of all positive cases [24]. ‘Survey’ apps gather information from door-to-door surveys, such as the Internal Survey App in Bhopal, developed by Accredited Social Health Activist (ASHA) to determine the primary contacts of the positive cases in areas where people could not afford a smartphone or internet [24]. In a heterogeneous country like India, where only 24% of citizens own smartphones [25], it is surprising that there were only three nation-wide, two state-wide, and four city-wide applications—within the 20 smart cities investigated—dedicated to collecting data through surveys.

The second broad functional category of COVID applications, namely ‘Special medical services’ (*n* = 66), is divided into six subcategories—‘Medical Information’, ‘Medical advice’, ‘Vaccination’, ‘Telemedicine’, ‘Helpline’, and ‘SOS’. ‘Medical Information’ apps provide information related to medical needs, such as hospital information, the number of beds and their availability, oxygen availability, etc. In Pune, the iHealWell App provided real-time updates on the treatment of critical patients. It was developed to inform patients’ family members about the treatment provided to them [26]. ‘Medical advice’ apps provide medical advice to people by professionals such as doctors, healthcare workers, or trained volunteers. For example, the Sehath Sathi App in Jaipur provided access to medical services such as consultation services, prescriptions, follow-ups, and delivery of medicines for both symptomatic patients and patients with other medical conditions [24]. ‘Vaccination’ apps allow booking for COVID vaccine shots, with the most famous one in this category being CoWIN which operated as the vaccination portal on a nationwide scale. CoWIN allowed citizens to schedule and register for vaccination, in addition to providing information on vaccination [27]. ‘Telemedicine’ apps are similar to ‘Medical advice’ apps, as they provide medical advice or consultations only through phone, allowing more accessibility to citizens who do not have access to smart technology or the internet. The control room in Bhopal engaged with a team of doctors who provided medical advice over the phone [28]. ‘Helpline’ apps, as the name suggests, are helpline services that direct people to various services, such as chennaicovidhelp.in in Chennai. Though initially launched to disseminate vital COVID-related information, it has since been upgraded to provide a database of mental health service providers, clinical guidance providers, COVID and oxygen helplines, etc., all verified and sourced from the government portal [29]. ‘SOS’ apps connect citizens directly to the Rapid Response teams to provide emergency services. The COVID-19 Safety App in Kochi had an SOS button to request immediate assistance in case of any emergency for those in quarantine [30].

The third broad functional category of COVID applications, namely, ‘Control and Monitoring’ (*n* = 60), is divided into four subcategories: ‘Self-assessment’, ‘Contact tracing’, ‘Tracking’, and ‘Monitor’. ‘Self-assessment’ apps guide people in conducting self-assessments at home, reducing the need to run to hospitals or clinics, thereby reducing overcrowding and further spreading of the virus. For example, in Belgaum, the AyushBelgavi App provided guidelines to conduct a self-assessment test to determine whether a person has been infected with the virus and thereby needs to self-isolate [24]. ‘Contact Tracing’ apps help trace the interactions or contacts of the people who tested positive. Aarogya Setu, widely known as the COVID App in India, is the best-known example requiring constant Bluetooth access and location tracking [21]. ‘Tracking’ apps are used to identify and mark hotspots by tracking the movement of people. For example, SMC COVID-tracker in Gujarat tracked the movement of home-quarantined people and alerted the authorities if they moved away from their respective homes with the help of the ‘Geo-fencing’ feature on the app [31]. ‘Monitor’ apps focus on individuals in home quarantine and send them alerts on breaking quarantine. Some ‘Monitor’ apps include mandatory self-check-ins using a selfie, such as Quarantine Watch. 

The final broad functional category of COVID applications, namely, ‘Other essential services’ (*n* = 52), is divided into six subcategories—‘Essentials’, ‘Travel’, ‘Donation’, ‘Volunteer’, ‘Grievances’, and ‘Other’. ‘Essentials’ apps provide essential goods and services, including emergency shelters. In Udaipur, the food control room was set up by the district government to deliver essential goods to citizens’ homes [32]. ‘Travel’ apps provide services related to travel. The Jeevan Seva App in New Delhi helped COVID patients and their families under home isolation to safely commute to hospitals or clinics through sanitized electric vehicles [33]. ‘Donation’ apps collect donations from citizens for COVID-related issues, such as Charity of Wheels in Jabalpur. It was an online portal in which the citizens could put in an online request to collect the donation. Then, vehicles would be sent to the doorstep to collect the donations, either in cash or as items [34]. ‘Volunteer’ apps allow citizens to register themselves to provide volunteering services during the pandemic. In Assam, Cova Assam (COVAAS) was launched by the Assam Government to assist in fighting the pandemic by providing additional hands [35]. ‘Grievances’ apps allow citizens to file complaints related to any issue linked to COVID, such as the Delhi Corona App in New Delhi. This app, in addition to providing information regarding the beds, also allowed one to file a complaint if any hospital refused to admit positive patients, for whom a remedy would be provided [36]. ‘Other’ apps provide other functions, including educational services, training medical staff and volunteers, internal communication between departments, over-the-counter medication tracking, services for migrant laborers, etc. In Karnataka, for example, the Tele ICU program provided training and guidance to the ICU staff [37]. 

In terms of technology dependency, the 125 COVID applications identified are either web-based or mobile-based (Table 3). The web-based applications (*n* = 48) only require internet access (IN), and not smartphones (e.g., SMC website and JSS COVID platform). In contrast, mobile-based applications include (1) SP—apps that require smartphone technology (*n* = 55) (e.g., COVID Sachetak and COVID Safety), and (2) X—apps that do not require direct access to smartphone technology or internet access (*n* = 22). Basically, in this second subcategory, third parties (e.g., volunteers and/or government employees) with smartphones (either in person or via helplines) collect data from people who do not have direct access to smartphones or internet access. Web-based applications commonly do not require registration or submission of personal information, reducing privacy concerns mostly raised in regard to mobile-based applications.

There was a range of controversies around COVID applications, such as the data privacy issues following data breaches in several cases. More specifically, the mandatory imposition of Aarogya Setu by the Home Ministry—with 100% coverage required in all workplaces—raised serious concerns [38], as no exceptions were provided for those without smartphones, and violation of this direction could lead to criminal penalties such as imprisonment. There was broad resistance by civil society involving 45 organizations and about 100 individuals against the mandatory nature of the national app [39]. As a result of this joint effort, the government changed its position on the compulsory use of the app to ‘best effort use’ [40].

Another specific example of the COVID application controversy in India related to CoWIN—the national vaccination portal—as it only operated in English even though only about 10% of the entire population in India speaks English [41]. CoWIN was also criticized for malfunctioning and crashing, which led to the development of a few state-level applications for vaccination (e.g., CoWIN Kar App in Karnataka and COVA Punjab) [42]. Indeed, proposals were made to develop a separate app for each state to provide a better vaccination interface [43].

Last but not least, the sheer number of COVID applications identified in the 20 smart cities raises questions about the level of collaboration (e.g., data exchange) and coordination among them—if any. While it seems that some local apps were developed in response to the shortcomings of the national apps (as in the case of CoWIN), questions remain about the efficiency of having to manage so many applications at times of crisis—considering that the majority of COVID applications were initiated by the government. 

#### 4.1.2. COVID-War-Rooms

To manage the spread of the virus, 47 of the 100 smart cities repurposed their pre-existing Smart City infrastructure—Integrated-Command-and-Control-Centers (ICCCs)—into COVID-War Rooms [6]. The COVID-War-Rooms provided a platform where government, health officials, and citizens (volunteers) came under one roof in the fight against COVID. The idea was that data harvested from various sources, such as COVID applications and spatial mapping analysis, would be consolidated in the form of a dynamic dashboard to enable informed decision making. Nevertheless, Oluoch [14] refers to COVID-War Rooms as a site of ‘doing things at a distance’, since the decisions are made from a remote distance without the decision makers experiencing the effects of the virus on the ground. 

We have identified 14 Integrated-Command-and Control-Centers (ICCCs) in the investigated 20 smart cities that had been built as part of the Smart Cities Mission, and then converted to COVID-War-Rooms in response to the pandemic (see Table 4). It is important to note that our desktop analysis of the functionalities of different COVID-War-Rooms heavily relies on a government report called “The Smart responses to COVID-19: A Documentation of Innovative actions by India’s Smart Cities during the Pandemic” [24]. We, however, have cross-examined the official narrative versus other publicly available online sources such as newspaper articles [44,45]. This is an area in which further empirical research is desperately needed to establish the ground truth of the operation details of the COVID-War-Rooms and the roles they played in response to COVID.

Table 4 summarizes the roles played by the different COVID-War-Rooms across five categories adopted from the framework developed by the Government of India [24]:

Tracking and Monitoring: As discussed earlier in the paper, COVID applications were developed to inform, monitor, control the virus, and provide essentials for citizens during the pandemic. Many smart cities used the ICCC-converted COVID-War-Rooms as a platform to gather the data received from COVID applications and other sources. This information was then transferred into dashboards, either GIS-based, Key Performance Indicator (KPI)-format, or both, to support informed decisions in response to the pandemic. For example, the COVID-19 Odisha dashboard (operated by the COVID-War-Room) provided information on the confirmed, active, recovered, and deceased cases in Odisha, different districts, and across India, spatially and graphically. It also provided hospital helpline numbers, an infection summary, the number of COVID hospitals in each district with their capacity, a testing summary, an age-wise and gender-wise summary, the state-wide vaccination coverage, and government orders and initiatives. In addition, it also allowed citizens to apply for CAPS—COVID-19 Ex-gratia Assistance Payment System, book an oxygen concentrator, and know the test status [45].

Out of the 14 ICCC-converted COVID-War-Rooms, 10 had a dashboard that was publicly accessible at the peak of the pandemic. 

Diagnosis: Medical officers were stationed at the COVID-War-Rooms providing telemedicine, including guiding self-assessment and counselling services, to citizens who did not have to go to hospitals or clinics, avoiding overcrowding such as in Bhopal [44]. Also, additional mobile clinics and door-to-door surveys were initiated to ensure early diagnosis. For example, in New Delhi, New Delhi Municipal Council Smart City Ltd. (NDMC SCL), along with New Delhi Municipal Council (NDMC), set up 24 × 7 flu centers away from the main hospitals to ensure the timely screening of patients. These centers were monitored from the COVID-War-Room through CCTV cameras [24]. 

Awareness and Capacity Building: To spread awareness among the citizens and dispel misinformation, smart cities used different platforms, such as public addressing systems, WhatsApp groups, and Instagram (monitored and controlled by the COVID-War-Rooms). Some cities such as Surat also focused on training frontline workers and volunteers to increase capacity building [24]. In addition, awareness campaigns were held in slums and social media; and mass communication methods were used to educate citizens—all coordinated and monitored by the COVID-War-Rooms. 

Sanitation: Considering the importance of sanitation and hygiene to prevent the rapid spread of the virus, various actions were taken by the authorities in regularly sanitizing the high-risk areas/hotspots and public spaces, including hospitals, the coordination and monitoring of which were carried out in the COVID-War-Rooms. In addition, the safe disposal of the waste generated from the treatment of COVID in hospitals was considered a priority. In Visakhapatnam, the Geographic Information System (GIS) layers generated in the COVID-War-Room included a spraying activity layer, allowing authorities to keep track of the areas that needed to be regularly sanitized [24].

Citizen Support: In addition to apps for essential services, some smart cities provided toll-free helplines to offer guidance and support for any COVID-related issues, managed by the COVID-War-Rooms. For example, Citizen Grievance Redressal Services (CCRS) was set up in the COVID-War-Room in Ahmedabad, comprising a team of 50 trained staff working around the clock, helping citizens resolve their grievances [24]. This also allowed for more localized actions catering to the different sections of society.

Table 4 has been colored-coded, with white indicating the actions led by the smart city commission, light grey indicating the actions solely led by the city administration, and dark grey indicating the actions led in collaboration. While the smart city commission played an essential role in fighting the pandemic in most cities, in Pune and Surat, the respective city administrations collaborated with the smart city commissions in leading the fight against COVID. The respective city administrations were responsible for almost all actions, while tracking and monitoring were conducted in collaboration. In Pune, while Pune Smart City Development Corporation Ltd. (PSCDCL) developed an integrated online dashboard, Pune Municipal Corporation (PMC) developed containment plans, and created a COVID response team, ward-wise primary contract tracing, and door-to-door surveys [24]. PMC was also taking the initiative in diagnostics by launching mobile dispensaries at the residents’ doorstep to ensure medical services and to provide medical and testing support in hotspot areas. It also collaborated with community groups to mobilize resources and food distribution. 

While most of the COVID-War-Rooms are no longer functional, a recent media article [46] reported that Bhubaneswar Municipal Corporation (BMC) had reactivated the COVID-War-Room, recalling the call center and re-engaging the doctors to tackle the latest wave, having had high COVID case numbers in the earlier waves. This indicates that the functioning of the COVID-War-Rooms needs further empirical investigation with specific attention to their effectiveness in a time of ongoing crisis.

#### 4.1.3. COVID Smart Public Spaces

COVID Smart public spaces refer to the use of smart technologies, such as drones, digital boards, and public addressing systems (PAS), in public spaces during the pandemic for sanitization, surveillance, and mass communication. The COVID smart public spaces identified in this paper are not comprehensive and/or all-inclusive across the investigated 20 smart cities; they are, however, spacious enough to establish a sense of diversity and prevalence of such practices, and to hopefully encourage further empirical research in this field.

There are multiple media articles [47,48] reporting the ways in which authorities used drones for different purposes such as delivery, surveillance, screening, and disinfection, making it feasible to carry out these tasks without any person-to-person contact. In cities such as Ludhiana and Jaipur, in addition to CCTV cameras, drones connected to the surveillance control rooms were used to monitor unwanted activity and the crowding of people (Figure 2a), and were accessible directly by police officers [24]. Vizzbee was one of the start-up incubators that developed drones to remotely monitor public spaces in Bhopal. In Bhubaneswar, the Bhubaneswar Municipal Corporation (BMC) used agriculture spray drones to spray sodium hypochlorite solution in high-risk public areas, such as bus stations and hospital premises, to sanitize and disinfect them [24]. In Chandigarh and Varanasi, Garuda Aerospace developed an Automated Disinfecting Unmanned Aerial Vehicle (UAV) named Corona Killer (Figure 2b) for the same purpose [49]. Drones were also used for mass communication to broadcast messages and create awareness among the people. In Vishakhapatnam, drones were used as digital boards for various messages controlled by COVID-War-Room [48]. Some drones were installed with temperature sensors, cameras, and GPS trackers to screen people from their homes in hot-spots requiring frequent monitoring [50]. 

In Solapur and Pune (Figure 2c), public addressing systems (PAS), such as travelling auto-rickshaws, were used to broadcast vital information [51]. In Belgaum and Davanagere, police personnel also used this system to issue warnings and alerts related to social distancing and overcrowding [24]. In addition to PAS, digital signboards/Variable Messaging displays (VMS) were used in cities such as Visakhapatnam and Kakinada for monitoring and communication [52]. 

While drones were possibly effective in surveillance during the pandemic, ensuring that people were following the lockdown and quarantine norms, the normalized use of these drones after the lockdowns has raised privacy concerns as they collect data in the form of videos and images [53]. In Kerala, the footage recorded in these surveillance drones is directly sent to the phones of police officers. The state has also hired private organizations to operate drones, which again questions the data protection and privacy of citizens due to the lack of transparency. Martin et al. [54], for example, argue that such rushed use of technologies can create serious security and safety concerns, especially when the countermeasures are not suitable or readily available.

#### 4.1.4. COVID Smart Subaltern Responses

The digital divide across India is massive, despite the ongoing government and non-government investment. This, combined with the fact that many official and top-down smart COVID responses failed to account for the vast linguistic diversity across the nation, limited their reach and impact. Nevertheless, since the beginning of the COVID pandemic, numerous media reports have shared the ways in which civil society in India joined forces (with each other but also with the government) to provide bottom-up support to the most vulnerable groups, filling the gaps in the top-down approach. We broadly categorize such bottom-up support as subaltern COVID responses. This is informed by existing research on subaltern urbanism [55], in which civic mobilization has happened by or for the subaltern population in response to the specific and intense COVID-related challenges in the subaltern spaces. Such media reports and the limited academic research available on the subject are complemented and further elaborated in our recent interviews with the relevant NGOs across India, as we heard from those who had worked in the subaltern spaces across the country long before the COVID crisis, and yet had to mobilize—on an unprecedented level in response to the pandemic—to fill the gaps in the official government-led COVID responses in defense of the subaltern population. Future research is needed to capture the diversity of subaltern voices to fully understand the scope and scale of their struggle during COVID, and the ways in which they used (mostly mundane) technologies to communicate, mobilize, and survive the crisis. Below, we offer a few examples of what we broadly identify as smart subaltern responses.

In Kochi, the district administration in association with local NGOs set up mobile clinic facilities to manage the spread of COVID among migrant laborers [24]. In Kerala, Kudumbashree—a self-help group run by women—set up community kitchens to distribute cooked food and food materials for those in need during the lockdown, including inter-state migrants and seniors [56]. While 10 community kitchens were set up in Kochi, 113 were set up in the Ernakulam district. 

Due to the restrictions imposed during the COVID lockdowns, the subaltern responses to COVID had to embrace mundane (widely available) technology-enabled smart communication platforms such as WhatsApp and Zoom. For instance, Mullick and Patnaik [25] describe how two mostly women-dominated workforces from ASHA (community health centers) and Anganwadi (neighborhood/rural childcare centers)—both severely impacted by COVID lockdowns—joined together by forming 190,000 WhatsApp groups and 2,200,000 neighborhood groups across the nation to support each other and other vulnerable people in need. Such massive bottom-up mobilization in mostly subaltern sectors and spaces then, in turn, attracted the attention of several state- and city-level governments, and relief packages, social security pensions, and interest-free loans were provided to these self-help groups. 

In our interviews, we also heard several stories of the ways in which different vulnerable groups across the nation used smart communication platforms (e.g., WhatsApp and Zoom) to organize and mobilize. One interesting case was about street vendors, who, in a sense, are the backbone of the informal economy across the nation. NGO representatives, interviewed by us, shared how surprised they were by the quick uptake of such technologies by the street vendors during the COVID lockdown. Such smart communication avenues were initially started as welfare checks with a large and yet vulnerable group, as many had lost their only source of income due to COVID restrictions. However, the online meetings soon turned into spaces to collect evidence about over-policing during COVID, as many street vendors providing essential services (e.g., food and vegetables) were unlawfully targeted by the police [57,58]. Then, in turn, partner NGOs with legal expertise were invited to such online meetings to provide basic legal advice to the essential worker street vendors, while other means of communication were highly limited. 

An important lesson emerging from our interviews with the NGOs active in the subaltern spaces across India is that the subaltern existence of smart responses was not limited to the duration of extended COVID lockdowns. If anything, our interviewees described an agile and responsive existence, as civil society had to fill the gaps in the otherwise mostly top-down approach to smart COVID responses every step of the way. For example, early vaccination is another important point. As pointed out earlier, CoWIN was the digital platform put forward to allow for early vaccination registration. However, the NGOs interviewed by us pointed out the limitations of the CoWIN as it only operated in English which restricted its reach. In response to such an oversight in the design of an official smart COVID response, the NGOs across the nation used their already established channels of communication with the most marginalized communities (e.g., mega WhatsApp groups set up with street vendors during the COVID lockdowns) to reach out to them and invite them to vaccination registration camps in nearby locations. We have heard inspiring stories of how the online reach out to street vendors and informal settlers led to arranging cheap modes of commute (e.g., buses) to move hundreds of thousands of the most vulnerable populations to early vaccination registration camps (and later to the actual vaccination clinics). 

## 5. Conclusions

This paper looks into the scope and scale of the smart responses to COVID in the first 20 cities prioritized for smart city implementation—as part of the Smart Cities Mission in India. It unravels the size and diversity within the smart COVID response, as 125 COVID applications, 14 COVID-War-Rooms, and numerous examples of smart public place initiatives are discussed. The findings include a typology of COVID applications, and shed light on the operations of COVID-War-Rooms throughout the nation. Most importantly, this paper puts forward the notion of subaltern smart COVID responses which emerged in response to COVID, as different sections of civil society across the nation joined forces—together and also with the government—in support of the subaltern population and spaces, filling the gaps in the otherwise mostly top-down approach to smart COVID responses. More empirical research is required to capture the diverse voices involved with COVID management and to fully recognize the scope and scale of the smart responses to COVID, in particular the subaltern smart responses. This paper also calls for future research to examine the hierarchy of urban systems to understand how COVID responses are laid out across the nation [19]. This is an important line of research, as it sheds light on the opportunities and challenges embedded in smart urbanism in response to crises more broadly—whether economic, environmental, or otherwise.

## Figures and Tables

**Figure 1 ijerph-20-07036-f001:**
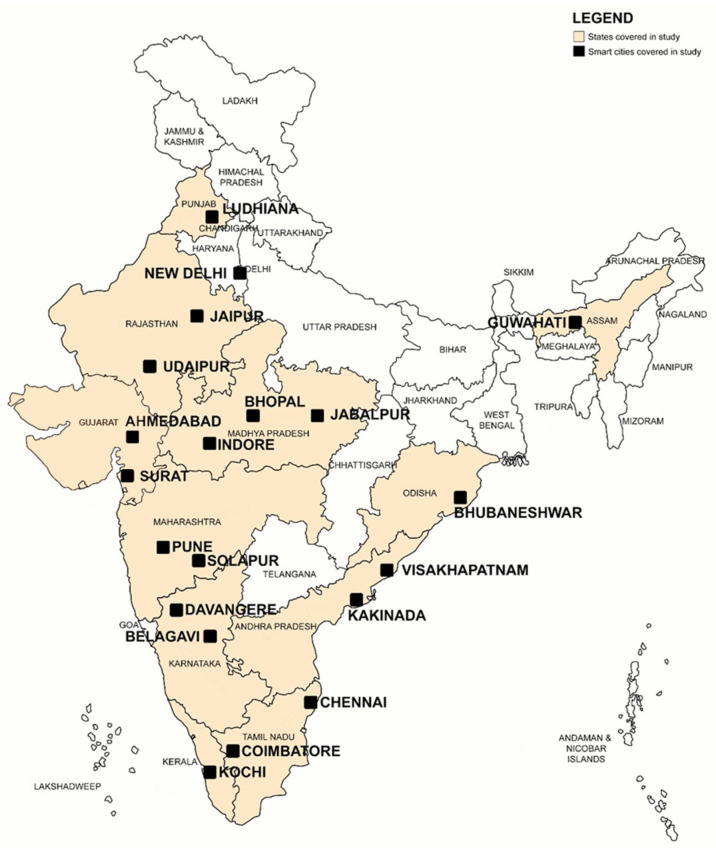
Map of the 20 smart cities investigated.

**Figure 2 ijerph-20-07036-f002:**
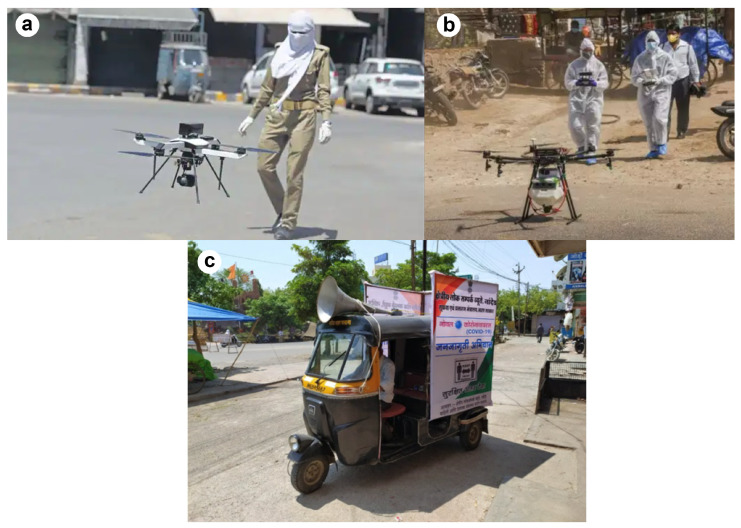
(**a**) Drones used for surveillance [47]; (**b**) drones used for sanitation [48]; (**c**) public addressing systems (PAS) [51].

**Table 1 ijerph-20-07036-t001:** Proposed approach of the study.

AIM: To Understand the Scope and Scale of the Smart Responses to COVID in India.
Desktop Analysis
Twenty cities prioritized for smart city implementation.Data sources: publicly available online sources, including (limited) academic research, official government websites and reports, media articles, social media posts, blogs, and app stores.
**Categories of CovTech**
COVID Applications(Based on scale, technology dependency, and functionality)	COVID-War-Rooms(Converted Integrated-Command-and-Control-Centres (ICCCs))	COVID Smart Public Place Initiatives(Not comprehensive. However, multiple examples indicate a pattern)	COVID Smart Subaltern Responses(Supported by interviews)
(To verify) 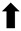
**Interviews with 20 NGOs**

**Table 2 ijerph-20-07036-t002:** COVID application categories by function and scale.

Function		Examples	Total Number of Apps			
Categories	Subcategories		Numbers	Scale		
				Nation-Wide	State-Wide	City-Wide
**General Information**	Information	Corona Safe Network, Kerala	80	7	21	31
	Geospatial data	JSS COVID platform, Jabalpur		2	3	7
	Survey	Internal Survey App, Bhopal		3	2	4
**Special medical services**	Medical Information	Ihealwell, Pune	66	2	6	7
	Medical Advice	Sehath Sathi, Jaipur		2	9	17
	Vaccination	CoWIN		2	2	1
	Telemedicine	Bhopal Control Room		0	0	3
	Helpline	COVID19 help, Chennai		0	5	6
	SOS	COVID safety, Kochi		0	1	3
**Control and Monitoring**	Self-assessment	AyushBelgavi, Belgaum	60	1	5	5
	Contact Tracing	Aarogya Setu		2	3	4
	Tracking	SMC COVID-19 Tracker		1	5	15
	Monitor	Quarantine Watch, Karnataka		2	5	12
**Other essential services**	Essential goods and services	Food control room, Udaipur	52	0	3	13
	Travel	Jeevan Seva, New Delhi		0	4	8
	Donation	Charity of wheels, Jabalpur		0	1	3
	Volunteer	COVAAS, Assam		0	2	1
	Grievances	Delhi Corona App, New Delhi		0	1	2
	Other	Tele ICU program, Karnataka		1	8	5

**Table 3 ijerph-20-07036-t003:** Categorization of COVID applications based on technology dependency across different scales.

Nation-Wide			State-Wide			City-Wide		
SP	IN	X	SP	IN	X	SP	IN	X
Arogya Setu	Just Dial	Ayush Sanjivini	COVID Odisha App	Odisha COVID Dashboard	Community Service Program	COVID Sachetak	pune.gov.in/corona-virus-updates (accessed on 15 November 2022)	Arogyadheer
COVID Kavach	Map my India		Corona Courage Odisha	MHCCMS	Gyan Brikshya	Tickme	SMC website	Helplines
Vee + Seva	My Gov		Mahakavach	COVID Suraksha	Aayu Sehat Sath	Saiyam	Amavad Mun. Co Website	COVID-19 Indore (internal App)
	Jaano		e-sanjivani OPD	Ghy Fights COVID	Karnataka Health Watch App	iHealwell	JSS COVID platform	door-to-door survey
	XACT		Covicare	Rajswasthya	Apthamitra	PMC home isolation	Charity of wheels	Aranya (internal)
	COVID-19 feedback		Coviguard	Shopsapp	QCIV	COVID info	RRT Application (internal app)	Helplines
	Cowin		COVAAS	covid.19.karnataka.gov.in (accessed on 15 November 2022)	Ghar Ghar Nigari	Sehath sathi	solapur.gov.in (accessed on 15 November 2022)	Surveys
	Migrant Workers resistance		Visit Assam	Yatri Karnataka Online Portal	CM Helpline	E-bazaar COVID-19	COVID-19 Tracker	Mobile Service
	COV India		Plassama App	Seva Sindhu Online portal		COVID Safety	devangere.nic.in (accessed on 15 November 2022)	Internal App
	Sahyog		Stay Home	Tele IC Program		Jabalpur Mart	Delhi Corona App	Food control room
			RajCovidInfo	KPME online portal		Jabalpur smart city surveillance	Delhi fights corona	Joint enforcement teams
			Rajnet	tnncovidbeds.tnega.org (accessed on 15 November 2022)		M Governance	Corona beds janta samvad	Udaipur COVID helpline
			SMC COVID-19 tracker	State portal for COVID monitoring		COC Platform	Migrant workers map	Psycho-social support
			Corona Safe Network			Cowin Solapur	COVID Status Dashboard	Survey (internal app)
			COVID-19 Jagratha			Control Room Dashboard	Google maps mapping	
			Gok Direct			Indore 311	ICCC dashboard	
			Blue Telemed			Jeevan seva app	GCC corona monitoring	
			Contact Tracing App			Kovai Care	tnepass.tnega (accessed on 15 November 2022)	
			Containment Watch			Ayush Belagavi	COVID-19 help chennai	
			Corona Watch App			Essential Service System	Tele-counseling center	
			Quarantine Watch App			Flyzy	ludhiana.nic.in (accessed on 15 November 2022)	
			Cowin kar App			GCC Vidmed	coronatraces.com (accessed on 15 November 2022)	
			COVA punjab			HBMS	Control Room Dashboard	
			COVID-19 Andhra Pradesh			Home Isolation Ludhiana		
			COVID Pharma			Itihas		
			Sarthak App			Nirmaya		
			MP COVID Response App					

**Table 4 ijerph-20-07036-t004:** A summary of the roles played by the COVID-War-Rooms.

Smart Cities	COVID-War-Rooms	Management	Actions					Dashboard
			Tracking and Monitoring	Sanitization	Awareness and Capacity Building	Citizen Support	Diagnosis	
Bhubaneswar	Intelligent City Operations and Management Center (ICOMC)	Bhubaneswar Smart City Limited (BSCL)		✓		✓		
Pune	Integrated Command and Control Center	Pune Smart City Development Corporation Ltd. (PSCDCL)	✓	✓	✓	✓	✓	✓
Jaipur	Abhay ICCC	Jaipur Smart City Limited (JSCL)	✓		✓			
Surat	Smart City Command & Control Center (ICCC)	Surat Smart City Development Limited (SSCDL)	✓	✓	✓	✓	✓	✓
Ahmedabad	Integrated Command and Control Center	Smart City Ahmedabad Development Limited (SCADL)	✓			✓		✓
Jabalpur	JSS COVID-19	Jabalpur Smart City Limited (JSCL)	✓				✓	✓
Visakhapatnam	City Operation Center (COC)	Visakhapatnam Smart City (VSC)	✓	✓	✓	✓		✓
Davanagere	Smart Command and Control Center (SCCC)	Davanagere Smart City Limited (DSCL)	✓		✓		✓	
New Delhi	Unified Command and Control Center	New Delhi Municipal Council Smart City Ltd. (NDMC SCL)	✓				✓	✓
Kakinada	Command and Communication Center (CCC)	Kakinada Smart City Corporation	✓	✓	✓	✓		✓
Belgaum	Integrated Command and Control Center	Belagavi Smart City Limited (BSCL)	✓		✓	✓		✓
Ludhiana	Central Command Center	Ludhiana Smart City Ltd. (LSCL)	✓			✓		✓
Bhopal	Integrated Command and Control Center	Bhopal Smart City Development Corporation Limited (BSCDCL)	✓	✓		✓	✓	✓
Total	14		12	6	7	9	6	10

Note: Background: the gradient shows the multiple roles played by the COVID-War-Rooms (high to low), with high being the darkest. ✓: function of the COVID-War-Rooms.

## Data Availability

Datasets used in this article are available on request from the corresponding author.

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
