# Peer review of "Smart Cities Mission in the Face of COVID: Scope and Scale of ‘Smart’ COVID Responses in India"

_ijerph, 2023, doi:10.3390/ijerph20227036_

Round 1

Reviewer 1 Report (Previous Reviewer 1)

Comments and Suggestions for Authors

The manuscript arrived at the level of acceptance, clearly research model, and well designed. I suggest creating a research framework diagram, for the authors.

Author Response

Thanks for the positive review. Reviewer 1 notes that the paper is "at the level of acceptance". So, I think that their suggestion to 'create a research framework diagram' shouldn't delay the acceptance of the paper. Nevertheless, we have already included 'Table 1. Proposed approach of the study' which is the same as 'the research framework diagram' Reviewer 1 is requesting.

Reviewer 2 Report (Previous Reviewer 3)

Comments and Suggestions for Authors

The authors addressed the main concerns and the revised version of the manuscript appears to be good. It looks ready for publication as far as I can tell.

Comments on the Quality of English Language

The English quality is acceptable.

Author Response

Thanks for the positive review. There is nothing to respond here.

Reviewer 3 Report (Previous Reviewer 2)

Comments and Suggestions for Authors

The manuscript has been substantially revised and is found to be in order.

Author Response

Thank you for the positive review. There is nothing to respond here.

This manuscript is a resubmission of an earlier submission. The following is a list of the peer review reports and author responses from that submission.

Round 1

Reviewer 1 Report

Comments and Suggestions for Authors

 This paper offers an analysis of responses to Covid in the f smart city implementation as part of the Smart Cities Mission in India. Although the paper has the potential, the work is perfectly acceptable. As it analyzes and unravels the smart Covid response, Covid applications, pandemic-related War-Rooms, and examples of smart public place initiatives. A few suggestions are attached in the review report. Also,

1. Please create a framework for your paper summary or the main points in a flowchart diagram. To improve the reader-friendly and better presentation. 

2. Future research section is not clear. Can be improved.

Comments on the Quality of English Language

Reviewer 2 Report

Comments and Suggestions for Authors

The authors have elaborated on the smart Govt measures taken to tackle COVID19 in India (20 smart cities). There are some concerns which must be addressed by the authors to improvise the quality of the manuscript.

1. Please incorporate manuscript organization in the end of Section 1.

2. Can the authors draw some contrast in the Smart COVID initiatives taken in other countries compared to India. This can incorporated in literature review. Comparisons can be tabulated if required.

3. Please make a block diagram in Methodology Section to showcase the proposed approach of this study. 

4. If possible, can the authors incorporate graphs to showcase the results at certain segments.

5. English is fine. Just can check the punctuations.

Comments on the Quality of English Language

Dear Editor,

I have highlighted my concerns towards this manuscript. 

Thank you for giving me an opportunity to review this manuscript.

With Regards,

Nishat

Reviewer 3 Report

Comments and Suggestions for Authors

Overall, the authors' research on scope and scale of smart Covid Responses in India is highly significant. The paper's methodology is appropriately selected, and the technical approach is relatively scientific, leading to compelling findings. The writing logic is clear and coherent. However, there are still some shortcomings that need to be addressed through revision and improvement.

1.        Please supplement the research review with studies on Smart Covid Responses from other countries worldwide, not limited to India.

2.        After the title of Section 4, it is recommended to add a paragraph or a framework diagram that provides an overview of the main content of this section to facilitate reader understanding.

3.        In section 4.1, the author categorizes Smart Covid Responses into four categories. Please provide further explanations for each classification.

4.        The subfigures in Figure 2 are not labeled as a, b, c.

5.        The manuscript still has some typos and requires careful checking and revision.

Comments on the Quality of English Language

Acceptable
